# BES1/BZR1 Family Transcription Factors Regulate Plant Development via Brassinosteroid-Dependent and Independent Pathways

**DOI:** 10.3390/ijms231710149

**Published:** 2022-09-05

**Authors:** Hongyong Shi, Xiaopeng Li, Minghui Lv, Jia Li

**Affiliations:** Guangdong Provincial Key Laboratory of Plant Adaptation and Molecular Design, School of Life Sciences, Guangzhou University, Guangzhou 510006, China

**Keywords:** brassinosteroid, brassinolide, transcription factor, BES1, BZR1

## Abstract

The BES1/BZR1 family is a plant-specific small group of transcription factors possessing a non-canonical bHLH domain. Genetic and biochemical analyses within the last two decades have demonstrated that members of this family are key transcription factors in regulating the expression of brassinosteroid (BR) response genes. Several recent genetic and evolutionary studies, however, have clearly indicated that the BES1/BZR1 family transcription factors also function in regulating several aspects of plant development via BR-independent pathways, suggesting they are not BR specific. In this review, we summarize our current understanding of this family of transcription factors, the mechanisms regulating their activities, DNA binding motifs, and target genes. We selectively discuss a number of their biological functions via BR-dependent and particularly independent pathways, which were recently revealed by loss-of-function genetic analyses. We also highlight a few possible future directions.

## 1. Introduction

Brassinosteroids (BRs) are an essential group of phytohormones regulating multiple physiological processes during plant growth and development. Within the past two and a half decades, the BR-signaling pathway has been gradually elucidated and has been considered as one of the best-characterized phytohormonal signaling pathways. In Arabidopsis, BRs are perceived by a plasma membrane-localized receptor–coreceptor complex, including the major receptor Brassinosteroid Insensitive 1 (BRI1), and its coreceptor, BRI1-Associated Kinase 1 (BAK1) [1,2,3,4,5]. Two BRI1-related receptor-like protein kinases (RLKs), BRI1-Like 1 (BRL1), and BRL3, also possess BR binding affinity but their roles are relatively weak [6,7]. BAK1 is a member of a subfamily of Leucine-Rich Repeat RLKs (LRR-RLKs), named Somatic Embryogenesis Receptor Kinases (SERKs). BAK1 was also designated as SERK3. There are in total five SERKs in Arabidopsis. At least three additional SERKs show functionally redundant roles with BAK1 in the BR signaling pathway, and BAK1 plays more significant role relative to its paralogs [3,5]. BR binding to the extracellular domain of BRI1 generates a new surface which facilitates its binding with the extracellular domain of BAK1, bringing the two intracellular kinase domains into close proximity, resulting in the phosphorylation and dissociation of a negative regulator, BRI1 Kinase Inhibitor 1 (BKI1) from the C-terminus of BRI1 and the transphosphorylation of the kinase domains of BRI1 and BAK1 [2,4,8,9,10]. The fully activated BRI1-BR-BAK1 complex triggers an intracellular phosphorylation cascade, inhibiting the activity and promoting the degradation of a GSK3-like kinase Brassinosteroid Insensitive 2 (BIN2), a key repressor of the BR signaling pathway [11]. When BR is absent, active BIN2 can phosphorylate two downstream transcription factors, bri1-EMS-Suppressor 1 (BES1) and Brassinazole Resistant 1 (BZR1) [12,13,14], inhibiting downstream BR outputs. Phosphorylated BES1 and BZR1 are less active due to decreased DNA-binding affinity and reduced protein accumulation in the nucleus [14,15,16,17]. BES1 and BZR1 can directly or indirectly regulate the expression of thousands of BR-responsive genes and ultimately affect plant growth, development, and stress adaptation [18,19,20,21]. Thus, BRs regulate cellular activities and biological processes via the BES1/BZR1 family transcription factors.

Interestingly, recent studies have also found that BES1/BZR1 family transcription factors are involved in regulating plant development via BR-independent pathways. It was found that BES1 can be activated by a signaling pathway involving a peptide ligand Tapetum Determinant 1 (TPD1) and its receptor Extra Microsporocytes (EMS1) during another development [22,23,24]. In addition, several other signaling pathways were found to use the BES1/BZR1 family to activate downstream response gene expression, such as TDR-TDIF, the heat stress resistance process, and Nematode-Induced LRR-RLK 1 (NILR1)-mediated signaling pathways [25,26,27]. Many previously published reviews have mainly focused on the signal transduction of BRs but not specifically on the family of BES1/BZR1. This review summarizes the current advances on the BES1/BZR1 family transcription factors, the mechanisms regulating their activities, DNA binding motifs, target genes, and their biological functions in regulating growth and development. We highlight a number of findings on the roles of the BES1/BZR1 family within the last a few years.

## 2. BES1/BZR1 Family in Plants

BES1 and BZR1 are two well-characterized transcription factors that have been originally identified through ethyl methane sulfonate (EMS)-based screens for *bri1-116* genetic suppressors and for mutants showing reduced sensitivity to a BR-specific biosynthetic inhibitor Brassinazole (BRZ), respectively [18,19]. In *Arabidopsis thaliana*, BES1 and BZR1 belong to a small protein group with six total members. The other four proteins in this family were designated as BES1-Homolog 1 (BEH1) to BES1-Homolog 4 (BEH4). BZR1 and BEH1 to BEH4 share 88%, 59%, 56%, 45%, and 48% amino acid sequence identity with BES1, respectively [28]. In a plant transcription factor database plantTFDB 5.0 (http://planttfdb.gao-lab.org/family.php?sp=Ath&fam=BES1; accessed on 1 June 2022), the six members of the BES1/BZR1 family were clustered into a single clade when the phylogenetic analysis was performed with full-length protein sequences [29]. Interestingly, two other proteins, BAM7 and BAM8, are also clustered with the six BES1/BZR1 family members. BAM7 and BAM8 can bind double-strand DNA (dsDNA) and show transcriptional activities since they contain a BES1/BZR1-type bHLH domain [30]. However, BAM7 and BAM8 also contain a glycoside hydrolase-like domain and share very low amino acid sequence identities with BES1 (12.6% and 10.1%, respectively). These two proteins are usually classified as members of the BAM (BETA-AMYLASE) family [31,32]. Therefore, it is generally thought that the BES1/BZR1 family only contains six members in Arabidopsis, BES1, BZR1, BEH1, BEH2, BEH3, and BEH4 (Figure 1).

The BES1/BZR1 protein family has mainly been studied in *Arabidopsis thaliana*, but its orthologs have been widely found in the plant kingdom [29,33,34]. Currently, homologous proteins have been identified and confirmed in a number of sequenced plant species, including *Oryza sativa*, *Zea mays*, *Solanum Lycopersicum*, *Nicotiana benthamiana*, *Medicago truncatula*, *Pyrus ussuriensis*, *Glycine max*, *Gossypium hirsutum*, *Brassica rapa*, *Triticum aestivum*, *Cucumis sativus*, *Beta vulgaris*, *Panax ginseng*, and *Marchantia polymorpha* [35,36,37,38,39,40,41,42,43,44,45,46,47,48,49,50,51]. It should be noted that some proteins are more closely related to BAMs (such as AtBAM7 and AtBAM8) within the evolutionary tree (Figure 1). The homologs of BES1 and BZR1 were discovered by genome and transcriptome sequencing in model or non-model plant species, whose biological roles need to be analyzed by genetic and biochemical approaches. For example, mutants of OsBZR1 from rice and GmBZL2, GmBZL3, and GmBEHL1 from soybean exhibit BR-related phenotypes [42,44,51,52,53]. It is not surprising that most orthologs in plants should function similarly to Arabidopsis BES1 and BZR1.

**Figure 1 ijms-23-10149-f001:**
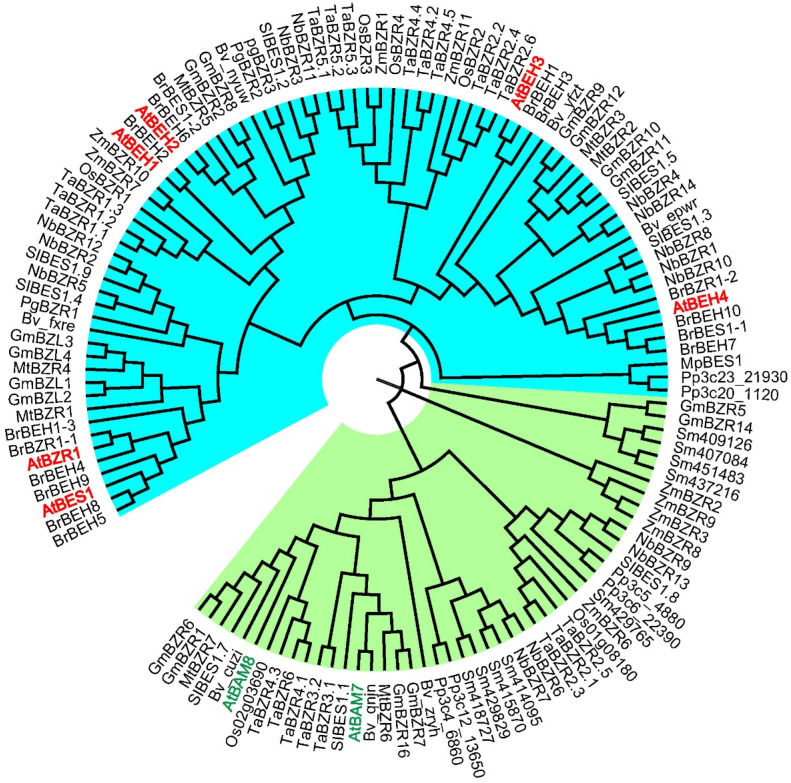
BES1/BZR1 family members and their phylogenic tree from *Arabidopsis* and a number of representative species. Phylogenetic relationships among BES1/BZR1 family proteins of the dicotyledonous plant species including *Arabidopsis thaliana* (At) [22,28,54,55,56], *Solanum Lycopersicum* (Sl) [45], *Nicotiana benthamiana* (Nb) [39], *Medicago truncatula* (Mt) [40], *Glycine max* (Gm) [44,52,53], *Brassica rapa* (Br) [43], *Triticum aestivum* (Ta) [36,38], and *Beta vulgaris* (Bv) [48]; monocotyledonous *Oryza sativa* (Os) [42,57], and *Zea mays* (Zm) [37,41]; and lower plant species *Selaginella moellendorffi* (Sm) [34,57,58], *Physcomitrella patens* (Pp) [27,34,57], and *Marchantia polymorpha* (Mp) [34]. BES1/BZR1 homologs are highlighted in the blue region and putative BAM homologs are highlighted in the green region. BES1/BZR1 family members from *Arabidopsis* are marked in red color. AtBAM7 and AtBAM8 are marked in green color. The phylogenetic tree was reconstructed using the neighbor-joining method. The full-length protein sequences for different species were obtained from the Ensembl database or corresponding references (Appendix A). The software MEGA-X was used to reconstruct the phylogenic tree, and FigTree (http://tree.bio.ed.ac.uk/software/figtree/, accessed on 1 June 2022) was used for visualization.

## 3. Activation of BES1 and BZR1 via Post-Translational Modifications

Protein phosphorylation is one of the common post-translational modifications of proteins, regulating their localization, function, and stability. BES1 and BZR1 are tightly regulated by the BR signaling cascade mainly via phosphorylation. When the BR level is low, BES1 and BZR1 are phosphorylated by BIN2 and its paralogs [13,14,59]. Recent studies have indicated that BES1 and BZR1 can be phosphorylated in both the cytoplasm and nucleus. When phosphorylated in the cytoplasm, the phosphorylated BES1 and BZR1 can be trapped in the cytoplasm by 14-3-3 proteins and degradation is promoted. When phosphorylated in the nucleus, on the other hand, the phosphorylated BES1 and BZR1 can be transported into cytoplasm, followed by interaction with 14-3-3 proteins and 26S proteasome-mediated degradation [17,60,61]. BIN2 phosphorylates BES1 and BZR1 on the serine (S) or threonine (T) residues of the consensus sequence S/TXXXS/T (X, any amino acids). There are about 25 putative BIN2 phosphorylation sites in BES1 and BZR1 [14,16,28]. In the presence of BR, BES1 and BZR1 are rapidly dephosphorylated by Protein Phosphatase 2A (PP2A) proteins [62]. It was found that BEH1, BEH2, BEH3, and BEH4 can be accumulated as their dephosphorylated forms in response to exogenous BR treatment, similarly to BES1 and BZR1 [28]. Consistently, OsBZR1, an ortholog of BZR1 in rice, can be phosphorylated by a GSK3-like kinase, OsGSK2. The phosphorylated OsBZR1 can be dephosphorylated by exogenous BR treatment [42,63,64,65]. These results indicate that reversible phosphorylation of the BES1/BZR1 family is an important and dynamic process in plants. The phosphorylation/dephosphorylation status of BES1/BZR1 family members determines their stability and transcriptional activity. Phosphorylated BES1 and BZR1 show low DNA binding capacity and their protein levels are reduced due to degradation by multiple ways [15,21,66,67,68,69]. The dephosphorylated forms of BES1 and BZR1 represent the active forms of the transcription factors.

Phosphorylation is not the only way to regulate the activity of BES1 and BZR1 activity. Recently, it was found that another form of post-translational modification, SUMOylation, can affect their functions [70,71,72]. SUMOylation is the process of SUMO proteins covalently linking to the lysine residues of their substrates, similar to ubiquitination. SUMOylation affects protein–protein interactions, subcellular localization, and the stability of target proteins in plants [73,74]. The SUMOylation of BES1 reduces its accumulation and inhibits its transcriptional activity [70]. An E3 ligase SIZ1 was found to SUMOylate BES1 at lysine 302 (K302), promoting its degradation. The mutated version BES1^K302R^ shows stronger transcriptional activity. The DNA binding ability of BES1 is increased in *siz1-2*. Differently from BES1, SUMOylated BZR1 can be accumulated in the nucleus due to its inaccessibility to BIN2, and it can promote growth, specifically under non-stressed conditions [72]. BZR1 can be SUMOylated at K280 and K310, overlapping with its BIN2 interaction motif, and thus blocking its interaction with BIN2. BZR1 can be deSUMOylated through a SUMO protease ULP1a. Low salt and high brassinolide (BL) promote the degradation of ULPa. In Arabidopsis, the SUMO sites found in BES1 and BZR1 are not all conserved among other members of this family. Therefore, whether SUMOylation is universal in BES1/BZR1 family members in regulating development and environmental adaptations remains to be elucidated.

BES1 and BZR1 also have oxidative modifications on cysteine residues upon hydrogen peroxide (H_2_O_2_) bursts, which can also affect transcriptional activity by enhancing the interaction with their partners, such as ARF6, PIF4, and SHR [75,76]. It was found that the oxidative modification of BZR1 does not affect its DNA binding ability or subcellular location. A cysteine mutation at position 84 in BES1 shows genetic similarity to a cysteine mutation at position 63 in BZR1, suggesting a conserved function of these two cysteine residues [75]. Amino acid sequence analysis has revealed that in Arabidopsis, the BZR1 cys84 is conserved with cys46 MpBES1 from *Marchantia polymorpha* [34], suggesting oxidative modifications of BES1/BZR1 family members are widely present in the plant kingdom.

Multiple post-translational modifications (PTM) were identified on BES1 and BZR1, altering their activities and functions in responding to BRs or environmental signals. SUMOylation does not affect BES1 phosphorylation, and phosphorylation does not affect SUMOylation [70]. Similarly, oxidative modifications do not affect the phosphorylation modification of BZR1 [75]. Plants use SUMOylation, phosphorylation, and oxidation to control the transcriptional activity of BES1 and BZR1 and presumably other members in this protein family. It is highly possible that different modifications can alter their downstream target gene expression profile, and therefore affect development or stress adaptation. However, the functions of these modifications in BR-independent pathways remain uncharacterized. It would be intriguing to investigate the relationship of various post-translation modifications and their corresponding signals, as well as the biological consequence of each modification.

## 4. DNA-Binding Motifs of the BES1/BZR1 Family

All six members of the BES1/BZR1 family in Arabidopsis contain a none-canonical plant-specific basic helix–loop–helix (bHLH) DNA binding domain (DBD) [28,58] (Figure 2A). DNA-binding transcription factors regulate gene expression by binding to their corresponding cis-regulatory elements in the promoter region. Hence, cis-regulatory elements play key roles in the transcriptional regulation of target genes [77,78]. By using a deoxyribonuclease I (DNase I) footprint assay and an in vivo chromatin immunoprecipitation (ChIP) analysis in Arabidopsis, He et al. identified that BZR1 can directly bind to the promoter regions of *CPD* and *DWF4*, containing a core sequence CGTGT/CG named BR response element (BRRE) [79]. Using an electrophoretic mobility shift assay (EMSA) and a ChIP approach, Yin et al. found that BES1 interacts with a different bHLH transcription factor BES1-Interacting MYC-Like 1 (BIM1). The complex binds to the E-box element (CANNTG, N is for any nucleotide) in the promoter region of *SAUR-AC1* [28]. Further studies combining ChIP with gene expression arrays (ChIP-chip) confirmed that BES1 and BZR1 regulate the expression of their target genes by binding to the E-box or BRRE in the promoter regions, respectively [18,19]. In addition, EMSA analysis showed that BEH3 and BEH4 can bind both the E-box and the BRRE sequences [80]. Therefore, BES1/BZR1 family proteins regulate target gene expression through binding to the cis-regulatory elements the E-box or BRRE.

In a yeast one-hybrid experiment, MpBES1, an ortholog of BES1 from mosses, was found to regulate the expression of Arabidopsis *CPD*, indicating that it can bind to the same regulatory element as Arabidopsis BES1. The E-box motif (CACGTG) was significantly enriched in the promoters of 500 top upregulated DEGs in *Mpbes1* [34]. In rice, OsBZR1 can directly bind to the E-box motifs of the *FC* promoter during tillering [42,63,64]. In pear (*Pyrus ussuriensis*), PuBZR1 was found to interact with the BRRE motifs in the promoter of its target gene *PuACO1* [49]. Therefore, the core DNA binding sequences of BES1/BZR1 family members are conserved among various plant species.

The crystal structure of BZR1-DBD with DNA has been resolved, and the DNA recognition mechanism of the BZR1 transcription factor has been established [58,81]. The BZR1-DBD complex forms a homodimer, and the N-terminal helix (helix I) of BZR1 is linked to major grooves on the dsDNA. Five BZR1 residues on helix I (Arg30, Arg34, Arg36, Arg40, and Arg52) and three residues (Lys61, His62, and Asn65) on the bHLH loop interact with the DNA phosphate groups. Two key residues, Glu37(i) and Arg41(i + 4), enable BRRE-core motif recognition [58]. BZR1-DBD is widely conserved in the genes of land plants, including ferns (*Selaginella moellendorffii*), and mosses (*Marchantia Physcomitrella*) (Figure 2B) [34,57,58]. For most orthologs, the amino acids involved in binding specificity and affinity are highly conserved (Figure 2B), suggesting that DNA recognition is carried out via a similar mechanism.

**Figure 2 ijms-23-10149-f002:**
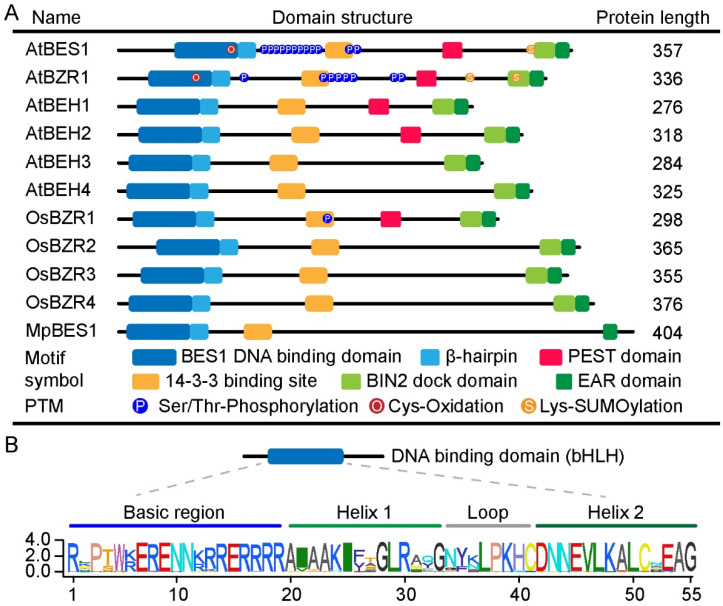
Conserved structural domains of BES1/BZR1 family proteins in Arabidopsis, rice, and *Marchantia polymorpha*. (**A**). The motif analysis was performed using MEME, and manual adjustments were carried out using the information from the literature. Motifs shown here include DNA binding domains and β-hairpins [58], 14-3-3 binding site [42,60], PEST domain [16,20,54], BIN2 docking domain [82], and EAR domain [83,84]. P shown in the diagrams represents the confirmed phosphorylation sites on serine or threonine residues [16,42,61,85], S represents SUMOylation on lysine residues [70,72] and O represents oxidation on cysteine residues [75]. (**B**). Consensus sequences of bHLH motifs in all the BES1 and BZR1 orthologs are shown in Figure 1. Multiple alignments were performed by Clustal Omega (Appendix A). SeqLogo in TBtools [86] was used to analyze the conserved sequence.

## 5. Other Conserved Motifs in the BES1/BZR1 Family

Other key structural motifs of BES1 transcription factors are shown in Figure 2A. Target proteins are interacted with by 14-3-3 proteins in a sequence-specific manner. The 14-3-3 binding motif is conserved and confirmed in BES1, BZR1, and OsBZR1. The cytoplasmic retention of 14-3-3 is essential for the inhibition of phosphorylated BES1 [42,60]. Additionally, PEST serves as a target sequence for protein degradation. In bzr1-1D, the Pro234 to Leu mutation stabilizes the protein, causing a constitutive BR response [54]. Likewise, in *bes1-D*, a Pro255 to Leu mutation was found in the PEST domain of BES1 [55]. As a result, bes1-D is more stable than BES1, and *bes1-D* exhibits a constitutive BR-activation phenotype [55]. The PEST domain, however, is not always conserved in the BES1/BZR1 family members. For example, a typical PEST domain was not found in BEH3, BEH4, and MpBES1 (Figure 2A) [34,80]. There is a 12-amino-acid BIN2-docking motif adjacent to the C-terminus of BZR1 that interacts directly with BIN2 [82]. There is an LxLxL type of EAR motif in the truncated region. In many cases, proteins containing the EAR motif can interact with co-repressor proteins such as Topless (TPL) [83,84,87,88]. In summary, domains found in BES1, except the DNA binding bHLH domain, are not always conserved in other paralogs or BES1/BZR1 family members from other plant species.

## 6. Target Genes of the BES1/BZR1 Family

Via binding to their regulatory sequences, transcription factors are able to regulate the expression of a large number of target genes. Identification of the target genes is crucial in understanding the biological functions of the BES1/BZR1 family. In Arabidopsis, BES1 and BZR1 have a relatively large number of direct target genes, including 3410 genes for BZR1 and 1609 genes for BES1 [18,19]. The target genes were identified via ChIP analysis and the gene expression profiles from two gain-of-function mutants, *bes1-D* and *bzr1-1D*. Comparative analysis of the target genes with BR-regulated genes revealed that BZR1 contains 953 BR-regulated genes (27.9%) among all of its 3410 target genes, and BES1 possesses 250 BR-regulated genes (11.6%) among all of its 1609 target genes. The direct target genes of other BEHs are generally unavailable [80,89,90,91]. Recently, transcriptome analysis of a hextuple/sextuple mutant *bzr-h* (*bzr1-4a bes1-4g beh1 beh2-4a beh3 beh4*) and the triple-null mutant of the BR receptors *bri-t* (*bri1 brl1 brl3*) revealed 1531 DEGs and 1410 DEGs in comparison to wild-type, respectively [56]. Overlap analysis has indicated that there are 534 shared DEGs between *bzr-h* (34.9%) and *bri-t* (37.9%). By comparing DEGs with All-BR-Regulated Genes (ABRG) from different resources, it was found that 1159 DEGs (75.7%) from *bzr-h* were BR-regulated genes.

It is noteworthy that there are still 372 DEGs (24.3%) in *bzr-h* that do not belong to the BR-regulated or BR receptors-regulated targets. Based on GO ontology, these genes are involved in multiple developmental processes, such as cell differentiation, meristem maintenance, reproduction, and flower development [56]. These results indicate that the BES1/BZR1 family regulates target genes in a brassinosteroid-dependent and brassinosteroid-independent manner.

## 7. Functions of BES1/BZR1 in Regulating Plant Growth and Development through a Canonical BRI1-BR-BAK1 Pathway

The identification and study of two gain-of-function mutants, *bes1-D* and *bzr1-1D*, provided valuable insights into our understanding of the functions of BES1 and BZR1. However, a single or double mutant of BES1 and BZR1 fails to show any characteristic BR-deficient mutant phenotypes [22,56,90,92], suggesting functional redundancy within the BES1/BZR1 family members. Consistently, a BES1 RNAi transgenic line showing reduced expression of *BES1*, *BZR1*, and several *BEHs* genes, displays a weak BR mutant-like phenotype [28,93]. Sextuple mutants of all BES1/BZR1 family members were then generated by a number of groups independently and they exhibit a more severe phenotype similar to the triple-null mutants of BR receptors [22,56]. These genetic results confirmed that the BES1/BZR1 family members are functionally redundant, and they play key roles in the BR signaling pathway.

BRs regulate a broad range of processes during plant growth and development, including hypocotyl elongation, root growth and development, photomorphogenesis, skotomorphogenesis, and reproduction [20,21,94,95,96,97,98,99]. In this section, we will mainly discuss several BR-dependent roles of the BES1/BZR1 family, which were mainly revealed via loss-of-function genetic analyses.

### 7.1. BES1 Regulates the Development of Female Gametophyte and Ovuled through a BR-Dependent Pathway

During female germline specification, BES1 and BZR1 control the division of megaspore mother cells (MMC) in plant ovules [100]. In the *bri1-116* and *qui-1* (*bes1-1 bzr1-1 beh1-1 beh3-1 beh4-1*) mutants, multiple MMC were identified in a single embryo sac of an ovule, which was not observed in the wild-type. This process is apparently regulated via the BR receptor and the downstream BES1/BZR1 family in sporophytic ovule cells. BES1 and BZR1 were also found to regulate ovule development in Arabidopsis. Outer integument cell length and cell numbers are significantly reduced in the ovules of *bri1-116*. These defects can be restored by a gain-of-mutation, *bzr1-1D* [101]. The hextuple mutant of the BES1/BZR1 family transcription factors, *bzr1-h*, shows similar outer integument growth defects.

### 7.2. BES1 Regulates Root Development through a BR-Dependent Pathway

Single-cell mRNA sequencing analyses of *bri1 brl1 brl3* and wild-type roots revealed that BR signaling is essential for precise cell division and cell expansion in root development [102]. Different tissues of a root contain different amounts of bioactive BRs [103], and an adequate amount of BR signaling is necessary for the development of a root, such as meristem division, and quiescent center (QC) maintenance [98,104]. In the vascular initials and QC cells, Brassinosteroids At Vascular And Organizing Center (BRAVO) is a specifically expressed R2R3-MYB-type transcription factor. BRs regulate the identity of QC through downregulating the transcription of *BRAVO* by BES1 as well as the heterodimerization of BRAVO [88]. *det2-1*, *bri1-116,* and *bzr-hw* (*bzr-h* weak allele) mutants display a reduced periclinal division and *bzr1-1D* shows an increased periclinal division. During root ground-tissue maturation, BZR1 and Short Root (SHR) form a transcriptional module that regulates periclinal division [76].

## 8. Functions of the BES1/BZR1 Family in Regulating Plant Growth and Development through a BR-Independent Pathway

As aforementioned, 372 DEGs (24.3%) identified in *bzr-h* are not contributed to by BRs [56]. Identifying BES1/BZR1-mediated but BR-independent signaling pathways is crucial for us in understanding the functions of the BES1/BZR1 transcription factors. This section mainly discusses what we have learned about BR-independent pathways involving in the tapetum and vascular development.

BES1/BZR1 orthologs are widely present in the plant kingdom, from lower to higher species. A loss-of-function mutant of MpBES1 in *Marchantia polymorpha* shows stunted thalli that cannot differentiate into adult tissues and reproductive organs [34]. From an evolutionary point of view, BES1/BZR1 family transcription factors play an ancestral role in the control of cell division and differentiation in plants. Because there are no BR receptors in *Marchantia polymorpha*, the observed functions of MpBES1 must be BR-independent.

### 8.1. BES1/BZR1 Family Regulates Tapetum Development

TPD1 is a secretory small peptide which specifically determines the formation of the tapetum [105,106]. TPD1 can bind to its receptor and co-receptor, EMS1 and SERK1/SERK2 [107,108,109]. The EMS1-TPD1-SERKs complex can regulate tapetal cell differentiation, but the molecular mechanism is not well understood. It was previously believed that BES1/BZR1 family members are mainly involved in regulating BR signaling. To investigate the genetic significance of BES1/BZR1 family members in the BR signaling pathway, a series of high order mutants were generated. Interestingly, quintuple mutants (*qui-1*, *bes1-1 bzr1-1 beh1-1 beh3-1 beh4-1*, and *qui-2*, *bes1-c1 bzr1-1 beh1-1 beh3-1 beh4-c1*) show a complete male-sterile phenotype due to a defective or missed tapetal cell layer in their anthers, respectively, suggesting that BES1/BZR1 family members are essential for tapetum development [22]. However, the BR-related mutants can produce pollen grains in their anthers, consistent with the results from a previous report [22]. In addition, the tapetal deficient phenotype of *tpd1* or *ems1* can be largely rescued by introducing the gain-of-function mutants *bes1-D* or *bzr1-1D* or both [22,23]. These genetic results suggest that the BES1/BZR1 family members regulate tapetum development, which is independent of the BR signaling pathway [22,23,56] (Figure 3).

Ectopic expression of *TPD1* and *EMS1* simultaneously in *bri1-116* leads to the accumulation of dephosphorylated BES1, similar to the outcome of BES1 upon the application of exogenous BRs [22,23]. More recently, EMS1 was found to interact with BSK1 and BSK3 [24]. It is likely that EMS1-TPD1-SERK1/SERK2 can regulate BES1 activity via the common downstream regulatory components of the BRI1-BR-BAK1 signaling cascade. Ultimately, activated BES1 can directly regulate the development of the tapetum by upregulating the expression of key genes such as *DYT1*, *SPL*, and *TDF1* [22,56].

### 8.2. BES1/BZR1 Family Functions in Vascular Development

In plants, vascular stem cells proliferate and differentiate into vascular tissues. The vascular stem cells are maintained by a peptide tracheary element differentiation inhibitory factor (TDIF), encoded by *CLE41* and *CLE44*, and its specific receptor Phloem Intercalated with Xylem (PXY), which is also known as the TDIF Receptor (TDR) [110,111]. GSK3-like kinases and BES1 function downstream from the PXY/TDR-TDIF signaling pathway that regulates xylem differentiation [25]. BIN2 and its paralogs can interact directly with TDR, and their activities can be regulated by TDR [25,112]. The nuclear-localized BZR1 in wild-type was decreased after TDIF treatment, which was not observed in the *tdr-1* mutant [25]. Constitutively active BES1 (*bes1-D*) can promote xylem differentiation [25]. Loss-of-function mutants of BZR1 or BES1 show reduced differentiation of the xylem [113]. The single-mutant phenotype of BZR1 or BES1 is weak, probably due to the functional redundancy of the BES1 paralogs.

In addition to xylem differentiation, BIN2-BES1 was also found to regulate cambial activity and promote phloem differentiation [113,114,115,116]. BIN2 and its paralogs negatively and redundantly regulate phloem differentiation. A GSK3 sextuple mutant shows more differentiated phloem cells, more sieve elements (SEs), and few companion cells (CCs). Conversely, GSK3 gain-of-function mutants show an increased number of CCs [117]. OPS and its homologs encode a polarly localized membrane-associated protein that positively regulates phloem differentiation [115,118,119]. OPS directly interacts with BIN2 and inhibits its activity [112]. Overexpression of *OPS* leads to the accumulation of dephosphorylated BES1, a phenotype similar to a brassinosteroid-overproduction phenotype. A loss-of-function mutant of *OPS*, *ops*, shows delayed phloem differentiation in the roots [114,118]. *bes1-D* and *bzr1-1D* can rescue phloem defects of *ops* in roots [115]. Because brassinolide (BL) application cannot alter the ratio of SE/CC in hypocotyl [117], BES1/BZR1 family-regulated phloem differentiation is likely via a BR-independent pathway (Figure 3).

The entire vascular volume in the sextuple mutant of the *BES1*/*BZR1* family *bes1-h* (*bes1-h*, *bes1-1 bzr1-2 beh1-2 beh2-5 beh3-3 beh4-1*) is significantly increased compared with that in wild-type [80]. The procambial cell layers are reduced, and the vascular loops are compressed in the *bes1-D* and *bzr1-1D* single and double mutants [80,115]. These results suggest that BES1/BZR1 family transcription factors function in maintaining vascular stem cells. It is worth noting that BES1/BZR1 family members play distinct roles in vascular cell differentiation. BEH3 plays a positive role in the maintenance of vascular stem cells, while BES1, BZR1, and BEH4 promoter vascular stem cell differentiation [80].

### 8.3. BES1 Function in Other Signaling Pathways

Another RLK, NILR1, is able to partially complement the phenotype of *bri1* when it is overexpressed, suggesting that the downstream components of BRI1 can be activated by NILR1 [27]. NILR1 was also named Germination Repression And Cell Expansion Receptor-Like Kinase (GRACE) [120]. Genetic and biochemical analyses showed that NILR1 and BRI1 share the co-receptor BAK1 and substrate BSKs and presumably other common downstream components. However, the detailed molecular mechanisms regulating the NILR1-BAK1-mediated signaling pathway are yet to be explored (Figure 3).

The BES1 protein can be activated via dephosphorylation by ABA-controlled PP2C in response to heat, and this process does not require the BR signaling pathway [26]. In BR signaling mutants such as *bri1-1*, *bin2-1*, and *det2-1*, BES1 can still be activated by dephosphorylation upon heat treatment. This process requires ABI1, a PP2C phosphatase. Heat stress tolerance is reduced in *bes1-1* and *bes1-2* but enhanced in *bes1-D*. These findings suggest BES1 can be activated via a BR-independent signaling pathway in regulating heat stress tolerance.

## 9. Conclusions and Perspective

BES1/BZR1 family transcription factors are widely present in the plant kingdom, including in ferns, mosses, monocotyledonous, and dicotyledonous plants. The activities of these proteins are largely controlled by a number of post-translational modifications, such as phosphorylation, SUMOylation, and oxidation. This family of transcription factors can bind to conserved DNA motifs in the promoter regions of their target genes. Target gene analysis by using transcriptome and ChIP-chip approaches have demonstrated that BES1 and BZR1 regulate the expression of their target genes through BR-dependent and BR-independent pathways. Therefore, BES1/BZR1 family members serve as hubs connecting different signal pathways (Figure 3). Different ligand–receptor pairs likely trigger the expression of different or partially different gene profiles via the BES1/BZR1 family members and result in distinct physiological processes.

Although it was reported that EMS1-TPD1-SERKs and NILR1-BAK1 signal perception complexes may share common downstream components with BRI1-BR-BAK1, such as BSKs and BIN2, to regulate the activities of BES1/BZR1 family transcription factors [22,23,24,27], it is quite possible that these transcription factors can also be activated via different mechanisms. An obvious question would be how the specificity was determined for different signal perception complexes by using same set of transcription factors. Although the BRI1-BR-BAK1 signal perception complex affects plant growth and development in a whole-plant level, many other signal perception complexes, on the other hand, are more tissue-specific. It was previously reported that BES1 and BZR1 can sometimes interact with different partners to trigger the expression of a specific set of genes [28,121]. Some partners may be expressed spatiotemporally. Another explanation is the functions of BES1/BZR1 family members are not completely the same, although they are largely overlapped. Different signal perception complexes may unequally activate different members of the BES1/BZR1 family.

Another question would be whether there are additional signal perception complexes which also use BES1/BZR1 family as their downstream transcription factors. Previous observations support this assumption. For example, in the *BES1* sextuple mutant, the entire anthers are not differentiated [22]. The female gametophyte is not developed. In addition, the roots are much shorter than that of the wild-type [56,80]. These results suggest there are additional signal perception complexes or pathways that also use BES1/BZR1 family members as their downstream regulatory components. Moreover, only seed plants possess BRI1-like receptors; lower plants such as liverworts do not contain BR receptors [34,122,123]. However, BES1 orthologs are found in both liverworts and seed plants, suggesting they were evolved much earlier than the BRI1-BR-BAK1 signaling pathway. These results suggest this family of transcription factors should have additional functions other than the BR-related roles. Further studies should concentrate on identifying these additional signaling pathways.

It should be noted that several photoreceptors, including UVR8, CRY1, CRY2, and phyB, can directly interact with BES1 or BZR1 to inhibit their DNA binding activity [124,125,126]. There are also some reports about the stability of BES1 and BZR1 [67,68,93]. These issues are out of the scope of this review. We are mainly focusing on discussing the signaling pathways using BES1/BZR1 as downstream transcription factors.

It has been demonstrated that BES1/BZR1 family transcription factors are directly relevant to a number of agronomically important traits in a few plant species. For instance, OsBZR1 controls lamina bending and grain filling in rice [120,127], and PuBZR1 suppresses fruit ripening in pears [49]. Future studies should also pursue their application in crop improvement via gene editing or molecular breeding.

## Figures and Tables

**Figure 3 ijms-23-10149-f003:**
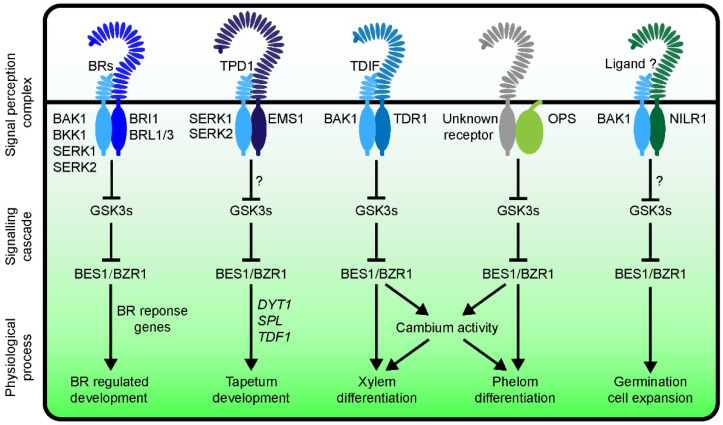
A current model showing BES1/BZR1 family members are involved in several signal pathways. In a classical BR signaling pathway, BRI1-BR-BAK1 regulates gene expression mainly via the BES1/BZR1 transcription factors family, which have been well characterized within the past two-and-a-half decades. Genetic and biochemical analyses indicated that the EMS1-TPD1-SERK1/SERK2 regulates tapetum development through activating BES1 and its paralogs. BES1 and BZR1 directly regulate the expression of tapetum developmental genes, such as *SPL*, *DYT1*, and *TDF1*. The TDR-TDIF-BAK1 signaling pathway regulates xylem cell differentiation via BES1/BZR1 family members. A plasma membrane-associated OPS regulates phloem differentiation via activating BES1. However, it is unclear whether there is an undefined RLK which can interact with OPS. NILR1 plays a role in seed germination and cell expansion. Co-receptor BAK1 is shared by NILR1, and downstream components may also be the same as those in the BRI1-BR-BAK1 signaling pathway. Question marks represent unconfirmed steps.

## Data Availability

Not applicable.

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
