# Peer review of "BES1/BZR1 Family Transcription Factors Regulate Plant Development via Brassinosteroid-Dependent and Independent Pathways"

_ijms, 2022, doi:10.3390/ijms231710149_

Round 1

Reviewer 1 Report

This manuscript reviewed the BES1/BZR1 family transcription factors, including their post-translational modifications and regulatory role in the transcriptional expression of downstream target genes via BR-dependent and independent pathways.

Generally:

BES1/BZR1 transcription factors play vital roles in plant growth and environment adaptability. Their regulation mechanisms have been well elucidated in genetic model plants such as Arabidopsis. The manuscript focused mainly on the functional role of BES1 and BZR1 as transcription factors but presented little information about the function of BES1 or BZR1 as regulatory hubs that integrate diverse signals to regulate plant growth and development and environmental responses. In addition, when reviewing the role of BES1 and BZR1 as transcription factors, the authors may briefly present the recent key progress on the molecular mechanisms of how BES1 and BZR1 regulate plant development and environment adaptability through their target genes. For instance, recent studies suggest that the receptor kinase OsWAK11 binds pectin and monitors pectin dynamics, which modulates the abundance of dephosphorylated BZR1 and thus the expression of the target genes, including those involved in cell wall remodeling.

Specifically:

(1)  A table could be included to summarize the post-translational modification (PTM) sites of BES1 and BZR1 and the responses of to PTM state to developmental and environmental cues.

(2)  PTM sites of BES1 and BZR1 could be presented in Figure 2A to show which domains of these PTM sites are localized.

(3)  Based on the above information, the authors may further review the mechanisms of how BES1 and BZR1 integrate different signals to regulate plant growth and environmental responses and discuss “why plants need to use SUMOylation, phosphorylation, and oxidation to control the transcriptional activity of BES1 and BZR1” (line 165-166).

(4)  Line 161, the key questions may include (i) how the PTM states of BES1 and BZR1 respond to developmental and environmental cues; (ii) what are the mechanistical and functional consequences of these modification events, and (iii) how these signaling events integrate diverse signals.

(5)  Italicize Arabidopsis thaliana in line 67 and other places.

(6)  Line 112, what does “its functional redundant proteins” mean here?

Author Response

This manuscript reviewed the BES1/BZR1 family transcription factors, including their post-translational modifications and regulatory role in the transcriptional expression of downstream target genes via BR-dependent and independent pathways.

Generally:

BES1/BZR1 transcription factors play vital roles in plant growth and environment adaptability. Their regulation mechanisms have been well elucidated in genetic model plants such as Arabidopsis. The manuscript focused mainly on the functional role of BES1 and BZR1 as transcription factors but presented little information about the function of BES1 or BZR1 as regulatory hubs that integrate diverse signals to regulate plant growth and development and environmental responses. In addition, when reviewing the role of BES1 and BZR1 as transcription factors, the authors may briefly present the recent key progress on the molecular mechanisms of how BES1 and BZR1 regulate plant development and environment adaptability through their target genes. For instance, recent studies suggest that the receptor kinase OsWAK11 binds pectin and monitors pectin dynamics, which modulates the abundance of dephosphorylated BZR1 and thus the expression of the target genes, including those involved in cell wall remodeling.

Response: Thanks for the suggestions. The main purpose of this review article is to point out that the activation of BES1/BZR1 could be through a number of BR-independent pathways.  Many people are confused by the BR signaling and BES1/BZR1 transcription factors. Although BES1/BZR1 family transcription factors are critical to the BR signaling pathway, they are not BR-specific transcription factors. Therefore, we did not complehensively discuss the functions of BES1/BZR1 which are known to be regulated by the BR signaling pathways. We only listed a number of currently revealed fuctions of BES/BZR1 family by loss-of-functional genetic analyses. It is hard to insert the information of OsWAK11 in this review article. We revised our title and abstract so that the scope of this review article is more clear.

1). A table could be included to summarize the post-translational modification (PTM) sites of BES1 and BZR1 and the responses of to PTM state to developmental and environmental cues.

Response: Thanks for the suggestion. We added PTMs of BES1/BZR1 in Figure 2. How PTMs are regulated by BR or other envioronmental cues are discussed in the many text. Therefore, we did not add the information in the table.

2). PTM sites of BES1 and BZR1 could be presented in Figure 2A to show which domains of these PTM sites are localized.

Response: We added the information in Figure 2A.

3). Based on the above information, the authors may further review the mechanisms of how BES1 and BZR1 integrate different signals to regulate plant growth and environmental responses and discuss “why plants need to use SUMOylation, phosphorylation, and oxidation to control the transcriptional activity of BES1 and BZR1” (line 165-166). Line 161, the key questions may include (i) how the PTM states of BES1 and BZR1 respond to developmental and environmental cues; (ii) what are the mechanistical and functional consequences of these modification events, and (iii) how these signaling events integrate diverse signals.

Response: We have modified our manuscript accordingly.

4). Italicize Arabidopsis thaliana in line 67 and other places.

Response: We modified the text in the revied manuscript.

5). Line 112, what does “its functional redundant proteins” mean here?

Response: We have changed to “its paralogs”.

Reviewer 2 Report

I checked your manuscript and described comments below.

The brassinosteroids are a kind of plant hormones and are very important because they are involved in the stem length and the differentiation of vascular bundles.

I think this paper describes the relationship between BES1/BZR1 family transcription factors and

brassinosteroid very well.

I suggest following point.

1.       I don't understand the reference for Figure 1. It is better to write the reference number.

2.       There is a description that Figure 1 was reconstructed. What kind of software did you use? It would be better to have a table of accession IDs for the amino acid sequences of the proteins used for this analysis.

3.       I would like a picture of the Clustal Omega results about Figure 2.

I don't think this paper has any major mistakes or grammatical problems.

Author Response

The brassinosteroids are a kind of plant hormones and are very important because they are involved in the stem length and the differentiation of vascular bundles. I think this paper describes the relationship between BES1/BZR1 family transcription factors and brassinosteroid very well.

1). I don't understand the reference for Figure 1. It is better to write the reference number.

Response: We added the references in the legend of Figure 1.

2). There is a description that Figure 1 was reconstructed. What kind of software did you use? It would be better to have a table of accession IDs for the amino acid sequences of the proteins used for this analysis.

Response: The software MEGA-X was used to reconstruct the evolutionary tree, and FigTree (http://tree.bio.ed.ac.uk/software/figtree/) was used for visualization. We modified the legend of Figure 1. Accession IDs for the amino acid sequences are provided in Supplementary Table 1 in the revised version.

3). I would like a picture of the Clustal Omega results about Figure 2.

Response: We have added the result of multiple sequence alignment in the Figure S1 of the revised version. Since the whole picture is too big, therefore, we added this data in Supplementary Figure S1.